# Novel Bifunctional Affibody Molecules with Specific Binding to Both EBV LMP1 and LMP2 for Targeted Therapy of Nasopharyngeal Carcinoma

**DOI:** 10.3390/ijms241210126

**Published:** 2023-06-14

**Authors:** Saidu Kamara, Yanru Guo, He Wen, Ying Liu, Lei Liu, Maolin Zheng, Jing Zhang, Luqi Zhou, Jun Chen, Shanli Zhu, Lifang Zhang

**Affiliations:** Institute of Molecular Virology and Immunology, Department of Microbiology and Immunology, School of Basic Medical Sciences, Wenzhou Medical University, Wenzhou 325035, China18647509030@163.com (Y.L.); 17835261229@163.com (M.Z.);

**Keywords:** affibody molecules, nasopharyngeal carcinoma, Epstein-Barr virus, LMP1-LMP2, targeted therapy

## Abstract

Antibodies are considered highly specific therapeutic agents in cancer medicines, and numerous formats have been developed. Among them, bispecific antibodies (BsAbs) have gained a lot of attention as a next-generation strategy for cancer therapy. However, poor tumor penetration is a major challenge because of their large size and thus contributes to suboptimal responses within cancer cells. On the other hand, affibody molecules are a new class of engineered affinity proteins and have achieved several promising results with their applications in molecular imaging diagnostics and targeted tumor therapy. In this study, an alternative format for bispecific molecules was constructed and investigated, named Z_LMP_110-277 and Z_LMP_277-110, that targets Epstein-Barr virus latent membrane protein 1 (LMP1) and latent membrane protein 2 (LMP2). Surface plasmon resonance (SPR), indirect immunofluorescence assay, co-immunoprecipitation, and near-infrared (NIR) imaging clearly demonstrated that Z_LMP_110-277 and Z_LMP_277-110 have good binding affinity and specificity for both LMP1 and LMP2 in vitro and in vivo. Moreover, Z_LMP_110-277 and Z_LMP_277-110, especially Z_LMP_277-110, significantly reduced the cell viability of C666-1 and CNE-2Z as compared to their monospecific counterparts. Z_LMP_110-277 and Z_LMP_277-110 could inhibit phosphorylation of proteins modulated by the MEK/ERK/p90RSK signaling pathway, ultimately leading to suppression of oncogene nuclear translocations. Furthermore, Z_LMP_110-277 and Z_LMP_277-110 showed significant antitumor efficacy in nasopharyngeal carcinoma-bearing nude mice. Overall, our results demonstrated that Z_LMP_110-277 and Z_LMP_277-110, especially Z_LMP_277-110, are promising novel prognostic indicators for molecular imaging and targeted tumor therapy of EBV-associated nasopharyngeal carcinoma.

## 1. Introduction

Epstein-Barr virus (EBV), a member of the herpesvirus family, has been reported to be associated with a number of malignancies, such as nasopharyngeal carcinoma (NPC), Burkitt’s lymphoma (BL), Hodgkin’s lymphoma, and gastric cancer [1]. In NPC, the latent membrane proteins LMP1 and LMP2 encoded by EBV are frequently detected and can play an important role in various biological processes, such as cell proliferation, cell migration, cell invasion, and apoptosis [2,3]. Moreover, studies have confirmed that both LMP1 and LMP2 can contribute to NPC pathogenesis [4,5], emphasizing the need to develop novel diagnostic and therapeutic approaches for EBV-associated NPC. Both LMP1 and LMP2 initiate and activate ERK-MAPK, JAK/STAT, NF-κB JNK/p38-SAPK, and PI3-K/Akt signal transduction pathways that affect undesirable phenotypic changes, including upregulation of c-Fos, c-Myc, and c-jun proto-oncogenes in NPC [6,7]. Consequently, monoclonal antibodies (mAbs) have been developed with the aim of blocking or inhibiting cellular signaling pathways in NPC [8,9,10]; however, sufficient efficacy has not been fully realized. More recently, there has been a growing interest in multi-targeted agents that substantially attenuate several cell signaling-related pathways in multiple cell types to achieve better single-agent efficacy and safety in a wider range of tumors [11]. This paradigm is illustrated by agents such as human epidermal growth factor receptor 3 (HER3) and other members of the epidermal growth factor receptor (EGFR) family [12,13,14,15,16,17], as well as several other agents currently under clinical development. For the application of effective multi-targeted agents in the clinic, it is imperative to understand the underlying mechanisms that influence their efficacy. Therefore, we generated novel bifunctional affibody molecules that bind simultaneously to two protein targets, EBV LMP1 and LMP2, for targeted treatment of NPC.

Bifunctional antibodies are designed to bind simultaneously to two different antigens and have been one of the most effective treatment breakthroughs in recent years [18]. They represent an emerging and exciting new area of cancer therapeutics and have shown efficacy in both preclinical and clinical trials [19]. In the past several years, the small molecule tyrosine kinase inhibitor (TKI) lapatinib, which selectively targets both EGFR1 and HER2, has proven to be an effective therapeutic strategy for HER-2-positive metastatic breast cancers [20], while bifunctional and bivalent antibodies, such as anti-HER2 and anti-EGFR1, showing high affinity, are currently being used in cancer diagnosis and treatment [21]. Although antibodies have been widely used in clinics, their large molecular size (150 kDa), multiple domains, limited tumor localization, and, at the same time, high production costs limit their application [22]. To circumvent the limitations, recombinant antibodies, including full-length antibodies, antibody fragments, antibody-enzyme fusion proteins, and more recently small antibody mimetics, have found increasing application in the diagnosis and treatment of cancer [23]. For example, Baeuerle et al. reported that bifunctional antibodies are suitable for targeted therapy through the use of a single-chain variable fragment (scFv) or CDR3 regions of monoclonal antibodies [24,25]. However, most of these small bivalent antibodies are unable to exhibit potent antitumor activity unless they are an effector moiety conjugated [26].

Affibody molecules are small (6.5 kDa) affinity proteins of 58 amino acids composed of a three-helix bundle and are potential substitutes for antibodies or scFvs in targeted tumor therapy [27,28]. Compared with antibodies, affibody molecules possess several advantages, such as faster penetration of tissue, high selectivity, and being easily obtained [29]. Due to their small size and ease of folding, affibody molecules can be made by peptide synthesis [30]. Moreover, rapid tumor penetration and clearance from non-specific sites make them useful for applications such as positron emission tomography (PET) imaging [31], optical and magnetic resonance imaging (MRI) [32], and fluorescence-guided surgery [33]. As of now, more than 400 published studies demonstrate that affibody molecules have been selected for a large variety of proteins with high affinity [34]. Affibody molecules target proteins such as epidermal growth factor receptor (EGFR) [32], human epidermal growth factor receptor 2 (HER2) [35], human epidermal growth factor receptor 3 (HER3) [36], vascular endothelial growth factor (VEGF) [37], human papilloma virus 16 oncoprotein E7 (HPV16E7) [38], latent membrane protein 2A-N terminal domain (LMP2A-N) [10], and latent membrane protein 1-C terminal domain (LMP1-C) [39]. Furthermore, dimeric HER2-specific affibody molecules and EGFR1/HER2 bispecific antibodies, respectively, inhibit the proliferation of ovarian cancer SKOV-3 cells [40,41]. To date, reported bivalent affibody molecules and nanobodies have become a promising tool for the diagnosis and therapy of diseases and are currently being used in cancer imaging [28,36]. However, there are no studies reported to fuse EBV LMP1 with LMP2 affibody molecules to form bifunctional molecules for dual targets and antitumor effects in mouse models, which inspired our group to design EBV LMP1-LMP2 bifunctional affibody molecules.

In our previous studies, we generated monospecific affibody molecules that target either LMP1 or LMP2 using phage display technology and explored the basis of their binding specificity in vitro and in vivo [10,39]. Based on these considerations, our group designed novel bifunctional affibody molecules for increased binding specificity and enhanced antitumor effects on NPC cells both in vitro and in vivo. We connected the LMP2A-N terminal affibody (Z_LMP2AN_110) to the LMP1-C terminal (Z_LMP1-C_277) using the (G4S)3 linker and further changed their positions to construct dual-affinity proteins, respectively, named Z_LMP_110-277 and Z_LMP_277-110. In this study, we constructed novel bifunctional affibody molecules (Z_LMP_110-277 and Z_LMP_277-110) with the potential for dual binding to the target proteins LMP1 and LMP2, antitumor effects on NPC cells, and in vivo evaluation of targeted therapy for NPC in xenograft tumor-bearing nude mice.

## 2. Results

### 2.1. Design and Construction of Bifunctional Affibody Molecules pET21a(+)/Z_LMP_110-277 and pET21a(+)/Z_LMP_277-110

The EBV Z_LMP1-C_277 and EBV Z_LMP2A-N_110 affibody molecules were selected using a phage display library that targets specific binders to the EBV LMP1-C (amino acids 186-387) and LMP2A-N (amino acids 1-119) proteins, respectively. The two monospecific affibody molecules (EBV Z_LMP1-C_277 and EBV Z_LMP2A-N_110) were connected by the linker (G4S)_3_ between Z_LMP1-C_277 and Z_LMP2A-N_110 and His-tag to produce Z_LMP_110-277 and Z_LMP_277-110 bifunctional affibody molecules (Figure 1A). The fusion linker provides spatial separation between two proteins while also improving biological activity and expanding expression yield [42]. Figure 1B shows a prediction of the three-dimensional structure of proteins using the SWISS-MODEL workspace (Figure 1B). This indicates that the connection between the two may not change their respective spatial conformations through flexible peptides.

DNA sequences encoding the bifunctional affibody molecules were cloned into *p*ET21a(+) to generate the recombinant plasmids *p*ET21a(+)/Z_LMP_110-277 and *p*ET21a(+)/Z_LMP_277-110 (Figure 1C). Then, the recombinant proteins were expressed in *E. coli* BL21 (DE3) after induction with 1 mM isopropyl β-D-1-thiogalactopyranoside (IPTG). Next, the His-tagged fusion proteins expressed were successfully purified by affinity chromatography using Ni-NTA resin and confirmed by sodium dodecyl sulfate-polyacrylamide gel electrophoresis (SDS-PAGE) analysis (Figure 1D). Western blotting further confirmed that fusion proteins were specifically recognized by the anti-His-tag mouse mAb (Figure 1E). Moreover, SDS-PAGE analysis showed that the final products were obtained with high purity (95%), which can then be used for future investigations.

### 2.2. Bifunctional Affibody Molecules Bind to Both LMP1 and LMP2 Simultaneously

In order to investigate whether the bifunctional affibody molecules were capable of binding to LMP1 and LMP2, surface plasmon resonance (SPR) was performed. The Z_LMP_110-277 and Z_LMP_277-110 were injected at different concentrations to flow over the sensor chip containing immobilized LMP1-C (amino acids 186-387) and LMP2A-N (amino acids 1-119) purified recombinant proteins (Appendix A). Sensograms were obtained after injection of affibody molecules (0.68 μM), and our results showed that bifunctional affibody molecules (Z_LMP_110-277 and Z_LMP_277-110) achieve good binding affinity and can simultaneously bind to both LMP1 and LMP2 proteins, respectively (Figure 2A,B). The dissociation equilibrium constant (KD) values of the binding affinity of Z_LMP_110-277, Z_LMP_277-110, Z_LMP1-C_277, Z_LMP2A-N_110, and SPA-Z scaffold (Z_WT_) to LMP1 are 3.28 × 10^−5^ mol/L, 8.08 × 10^−7^ mol/L, 3.56 × 10^−6^ mol/L, 1.96 × 10^−4^ mol/L, and 1.35 × 10^−1^ mol/L, respectively (Table 1). The KD values of Z_LMP_110-277, Z_LMP_277-110, Z_LMP1-C_277, Z_LMP2A-N_110, and SPA-Z scaffold (Z_WT_) to LMP2 are 1.04 × 10^−5^ mol/L, 1.92 × 10^−6^ mol/L, 6.47 × 10^−4^ mol/L, 3.92 × 10^−6^ mol/L, and 1.33 × 10^−1^ mol/L, respectively (Table 2). When comparing the binding affinity of bifunctional affibody molecules (Z_LMP_110-277 and Z_LMP_277-110) and monospecific affibody molecules (Z_LMP1-C_277 and Z_LMP2A-N_110) to LMP1 and LMP2, the KD values showed similar binding profiles; however, Z_LMP_110-277 and Z_LMP_277-110 simultaneously bind to both LMP1 and LMP2 compared to Z_LMP1-C_277 and Z_LMP2A-N_110. As expected, there was no interaction between the SPA-Z scaffold (Z_WT_) and LMP1 and LMP2.

### 2.3. Analysis of the Binding Selectivity of Bifunctional Affibody Molecules to Cells Expressing LMP1 and LMP2

To measure the expression levels of LMP1 and LMP2 in NPC-positive cells (C666-1 and CNE-2Z) and NPC-negative cells (HNE-2), quantitative reverse transcription polymerase chain reaction (qRT-PCR) and Western blotting assays were performed. The results of qRT-PCR analysis showed that LMP1 and LMP2 were highly expressed in NPC-positive cell lines compared to an NPC-negative cell line (Figure 3A,C). The qRT-PCR results were further confirmed by Western blotting assays (Figure 3B,D). Next, we study the cellular binding of Z_LMP_110-277 and Z_LMP_277-110 to NPC-positive cells using an indirect immunofluorescence assay. Our results showed that bright green fluorescence appeared at the juxtamembrane area of C666-1 and CNE-2Z that were treated with bifunctional or monospecific affibody molecules, as well as higher fluorescence intensity in cells treated with Z_LMP_110-277 or Z_LMP_277-110 compared to Z_LMP1-C_277 and Z_LMP2A-N_110 (Figure 3E,F). However, NPC-positive cells treated with SPA-Z scaffold (Z_WT_) and HNE-2 treated with bifunctional or monospecific affibody molecules did not exhibit any visible fluorescence signal (Figure 3G).

In addition, confocal immunofluorescence and co-immunoprecipitation (Co-IP) assays were performed to study intracellular protein-protein interactions. As shown in Figure 4A,B, bifunctional affibody molecules (green) can interact with both targets LMP1 and LMP2 (red) and colocalize (yellow). By comparison, monospecific affibody molecules can only interact with their individual target proteins, LMP1 or LMP2. SPA-Z scaffold (Z_WT_) did not display any obvious fluorescence signals. The fluorescence intensity of C666-1 incubated with Z_LMP_277-110 was higher than that incubated with Z_LMP_110-277 (Appendix A). Furthermore, Co-IP assays add more evidence that Z_LMP_110-277 and Z_LMP_277-110 interact with LMP1 and LMP2 proteins (Figure 4E,F). Nonetheless, Z_LMP1-C_277 and Z_LMP2A-N_110 only interact with LMP1 or LMP2 (Figure 4C,D). Taken together, these results suggest that Z_LMP_110-277 and Z_LMP_277-110 can recognize and bind simultaneously to both LMP1 and LMP2 proteins.

### 2.4. In Vivo Tumor Target Efficacy of Z_LMP_110-277 and Z_LMP_277-110

Furthermore, the in vivo biodistribution and tumor-targeted ability of Z_LMP_110-277 and Z_LMP_277-110 were investigated in nude mice bearing NPC-positive cell lines (C666-1 and CNE-2Z) and NPC-negative cell lines (HNE-2) tumor xenografts. The nude mice bearing tumor cell xenografts were intravenously injected with Dylight-755-labeled bifunctional or monospecific affibody molecules of 100 μg in 150 μL phosphate buffered saline (PBS) and then observed with a near-infrared imaging system at different time points after tail vein injection. As shown in Appendix A, Dylight-755-labeled bifunctional and monospecific affibody molecules were widely distributed throughout the body within 0.5 h after injection, excreted from the kidneys, and cleared from the body within 48 h. Then, we investigated the tumor-targeted ability of Dylight-755-labeled Z_LMP_110-277, Dylight-755-labeled Z_LMP_277-110, Dylight-755-labeled Z_LMP1-C_277, and Dylight-755-labeled Z_LMP2A-N_110 in a tumor-bearing mouse model. In NPC-positive cell (C666-1 and CNE-2Z) xenograft models, we observed that a strong fluorescence signal was detected at the xenograft tumor site 1 h post-injection of Dylight-755-labeled bifunctional and monospecific affibody molecules, peaked at 4 h, and remained for 24 h (Figure 5A,B). Nevertheless, no tumor-specific fluorescence signal was detected post-injection of Dylight-755-labeled bifunctional and monospecific affibody molecules in the HNE-2 xenograft model (Figure 5C). Moreover, the Dylight-755-labeled SPA-Z scaffold (Z_WT_) affibody molecules did not show any tumor-specific signal in xenograft models. These results showed that Z_LMP_110-277 and Z_LMP_277-110 bifunctional affibody molecules could accumulate at tumor locations in NPC-bearing mice with high specificity in vivo.

### 2.5. In Vitro Efficacy of Bifunctional Affibody Molecules

Next, to determine the antitumor efficacy of Z_LMP_110-277 and Z_LMP_277-110, cell viability assays were performed using C666-1, CNE-2Z, and HNE-2 cell lines. These cell lines were incubated for 72 h with increasing concentrations of Z_LMP_110-277, Z_LMP_277-110, Z_LMP1-C_277, and Z_LMP2A-N_110. After treatment with bifunctional or monospecific affibody molecules, the cell viability of C666-1 and CNE-2Z cells was reduced in a dose- and time-dependent manner (Appendix A). The half maximal inhibitory concentration (IC50) values of Z_LMP_110-277, Z_LMP_277-110, Z_LMP1-C_277, and Z_LMP2A-N_110 in C666-1 were 3.375 ± 0.180 μM, 3.265 ± 0.182 μM, 4.041 ± 0.296 μM, and 4.611 ± 0.271 μM, respectively. IC50 values in CNE-2Z were 3.245 ± 0.458 μM, 2.545 ± 0.182 μM, 7.021 ± 0.953 μM, and 3.556 ± 0.174 μM, respectively. The bifunctional affibody molecules showed no obvious inhibitory effect on HNE cells, whereas the SPA-Z scaffold (Z_WT_) had no effect on any of the three cell lines used in this study. The IC50 value of 10 μM was selected for further studies, and then we evaluated the efficacy of Z_LMP_110-277, Z_LMP_277-110, Z_LMP1-C_277, Z_LMP2A-N_110, and cisplatin (positive control) over the course of 12, 24, 36, 48, and 72 h. The Z_LMP_110-277, Z_LMP_277-110, Z_LMP1-C_277, and Z_LMP2A-N_110 affibody molecules reduced the cell viability of the C666-1 and CNE-2Z cell lines (Figure 6A). In addition, colony formation assays were used to test the long-term effects of Z_LMP_110-277 and Z_LMP_277-110 on NPC-positive cell proliferation. As reported in Figure 6B,C, long-term treatments of C666-1 and CNE-2Z with bifunctional affibody molecules significantly inhibited colony formation in NPC-positive cell lines compared to NPC-negative HNE-2. More importantly, Z_LMP_110-277 and Z_LMP_277-110, especially Z_LMP_277-110, exhibited higher antitumor effects than Z_LMP1-C_277 and Z_LMP2A-N_110 in both NPC-positive cell lines.

To further investigate the tumor-suppressive effects of Z_LMP_110-277 and Z_LMP_277-110 in C666-1 and CNE-2Z, flow cytometry was employed. Treatment of NPC-positive cells with bifunctional or monospecific affibody molecules increased the percentage of cells in the G0/G1 phase but reduced the percentage of cells in both the S and G2/M phases compared to the NPC-negative cells (Figure 6D,E). Taken together, these results suggest that bifunctional affibody molecules (Z_LMP_110-277 and Z_LMP_277-110) significantly suppressed NPC cell proliferation compared to their monospecific counterparts (Z_LMP1-C_277 and Z_LMP2A-N_110) and had no antitumor effect on HNE-2 cells.

### 2.6. Downregulation of the MEK/ERK/p90RSK Signal Transduction Pathway by Bifunctional Affibody Molecules in NPC Cells

The expression of LMP1 and LMP2 contributes to tumor cell proliferation, survival, motility, and invasion [2,3] and is mediated by several signal transduction pathways. Raf-MEK-ERK activates the 90 kDa ribosomal S6 kinase (p90RSK), resulting in increased transcription factors [43]. To explore the effects of bifunctional affibody molecules on the MEK/ERK/p90RSK pathway, Western blotting was performed. As shown in Figure 7A,B, there is a decrease in the level of phospho-Raf-1^(Ser338)^ in a concentration- and time-dependent manner after treatment with Z_LMP_277-110 compared to the control groups mock and SPA-Z scaffold (Z_WT_). As stated by the above result, 10 µM was selected for further study of the downstream targets. Our results showed decreased expression levels of phospho-MEK1/2^(Ser217/Ser221)^, phospho-ERK1/2^(Thr202/Thr204)^, phospho-p90RSK^(Ser380)^, and the transcription factors (c-Fos and c-Myc) after treatment with Z_LMP_110-277, Z_LMP_277-110, Z_LMP1-C_277, and Z_LMP2A-N_110 in NPC-positive cell lines (Figure 7C,D). Moreover, bifunctional affibody molecules had more pronounced effects on the MEK/ERK/p90RSK signaling pathway than monospecific affibody molecules (Figure 7E,F). Figure 7G shows a schematic illustration of bifunctional affibody molecule downregulation of the MEK/ERK/p90RSK pathway in NPC cells. Taken together, our results further confirmed that bifunctional affibody molecules significantly inhibit the proliferation and progression of NPC cells.

### 2.7. In Vivo Therapeutic Efficacy of Bifunctional Affibody Molecules

In vivo therapeutic efficacy of Z_LMP_110-277, Z_LMP_277-110, Z_LMP1-C_277, and Z_LMP2AN_110 was evaluated in C666-1, CNE-2Z, and HNE-2-bearing nude mice by measuring tumor growth in mice. Tumors grew much faster in groups treated with PBS and SPA-Z scaffold (Z_WT_) than Z_LMP_110-277, Z_LMP_277-110, Z_LMP1-C_277, Z_LMP2AN_110, and cisplatin (Figure 8A–D). More importantly, treatment with bifunctional affibody molecules strongly decreased tumor growth in NPC-bearing nude mice after 15 days of treatment compared to monospecific affibody molecules. Nevertheless, Z_LMP_110-277 and Z_LMP_277-110 did not exhibit any inhibitory effects on HNE-2 tumor growth (Figure 8E,F). Altogether, these results support the idea that bifunctional affibody molecules Z_LMP_110-277 and Z_LMP_277-110, especially Z_LMP_277-110, are promising anti-cancer agents for EBV-associated NPC.

## 3. Discussion

In recent years, researchers have focused on new therapy approaches to enhance the efficacy of drugs on tumor cells as compared to conventional cancer treatments. Current clinical practice utilizes mAbs extensively as treatment options for several cancer types, such as ovarian cancer, breast cancer, B-cell non-Hodgkin’s lymphoma, etc. [44,45,46]. Moreover, blinatumomab, an anti-CD19/CD3 bispecific antibody, showed great therapeutic efficacy with a good safety profile in lymphoid leukemia patients [47]. However, there are many pitfalls associated with mAbs and bispecific antibodies, such as limited depth of tumor penetration, potential mechanisms of resistance, and high manufacturing costs. Therefore, new strategies for the development of molecularly targeted tumor therapy are urgently needed.

Since affibody molecules were introduced three decades ago as antibody mimetics, a number of them have been generated and characterized. More recently, an affibody molecule against IL-17 (ABY-035) has entered clinical study and been demonstrated to be safe and tolerable [29]. So far, numerous bioengineered protein probes (scFv, Fv, and Fab) with smaller size, shorter circulation time, and deep tumor penetration have been used in a variety of medical applications, including blocking protein-protein interaction, targeted delivery of payload, inhibitory effect of peptide aggregation, and molecular imaging diagnosis [48,49]. In this study, we generated two novel bifunctional affibody molecules (Z_LMP_110-277 and Z_LMP_277-110) capable of simultaneously binding to LMP1 and LMP2, which display oncogenic properties and induce phenotypic alteration in epithelial cells. Then, we successfully produced these bifunctional affibody molecules using a prokaryotic expression system and evaluated their binding affinity and specificity to LMP1 and LMP2. The bifunctional affibody molecules showed binding to both LMP1 and LMP2 as compared to their monospecific counterparts. These results were similar to previous published studies from our laboratory [50,51]. In addition, immunofluorescence co-localization studies demonstrated that bifunctional affibody molecules (Z_LMP_110-277 and Z_LMP_277-110) could bind specifically to NPC-positive cell lines (C666-1 and CNE-2Z), as well as colocalize with both LMP1 and LMP2 proteins. Moreover, co-immunoprecipitation provided more evidence of the protein-protein interactions of bifunctional affibody molecules with LMP1 and LMP2 proteins. A study with radiolabeled affibody Z_HER2:342_ demonstrated fast tumor uptake in xenograft models and provided a higher imaging-specific contrast in HER2-positive SKOV-3 xenografts than the HER2 scFv antibody fragment [35]. By contrast, we also noticed similar data that Dylight-755-labeled bifunctional affibody molecules (Z_LMP_110-277 and Z_LMP_277-110) could rapidly accumulate in tumor locations in NPC-positive xenograft models. Our results suggest that Z_LMP_110-277 and Z_LMP_277-110 have good binding affinity and specificity to LMP1 and LMP2 both in vitro and in vivo and also solidify the potential of using Z_LMP_110-277 and Z_LMP_277-110 as molecular imaging probes.

NPC is a tumor of the head and neck that is frequently caused by the Epstein-Barr virus [1]. Despite recent advances in mAbs, radiotherapy, and surgery, overall patient cure is achieved in less than 50% of cases [8]. In addition, distant metastasis is the leading cause of treatment failure in NPC, with a median survival of 10 months or less [52]. Concurrent cisplatin chemoradiotherapy is the standard of care for locally advanced NPC [53], but it is associated with significant toxicity in NPC patients [54]. Therefore, new treatment agents that have the potential to improve NPC patient outcomes and that show a reduced toxicity profile are desperately needed. There are a variety of potential mechanisms for the use of mAbs in the treatment of cancer. For example, antibodies may target a specific portion of tumor development, such as growth factor receptors or receptor-ligand interactions, to kill tumor cells using death-receptor-mediated pathways [55,56]. Moreover, antibodies are capable of inducing complement-mediated cytotoxicity (CDC) or antibody-dependent cellular cytotoxicity (ADCC) [57]. Nonetheless, some of these antibodies were sometimes unable to stimulate antibody-mediated immune responses, including CDC and ADCC, which are important for destroying malignant tumor cells. In order to overcome this problem, fragment variable (Fv), single-chain Fv antibody fragments (scFvs), fragment antigen binding (Fab), and affibody molecules have been successfully used in cancer therapy and targeted drug delivery [58] and have been confirmed to target tumor-specific antigens with therapeutic effects. Among others, affibody molecules can be engineered to form dimers and trimers by varying the length of their peptide linkers, which can effectively improve binding affinity to target proteins [51,59,60]. In vitro cell viability, colony formation, and flow cytometry assays indicated that treatment with bifunctional affibody molecules (Z_LMP_110-277 and Z_LMP_277-110) inhibited the growth of NPC cell lines (C666-1 and CNE-2Z) and were non-cytotoxic toward the NPC-negative cell line (HNE-2). Importantly, the Z_LMP_110-277 and Z_LMP_277-110, especially Z_LMP_277-110, were more potent than either of the monospecific affibody controls (Z_LMP1-C_277 and Z_LMP2A-N_110) in NPC-positive cells. Furthermore, in vivo evaluation of antitumor therapeutic efficacy showed that the antitumor activities of Z_LMP_110-277 and Z_LMP_277-110 were higher than those of Z_LMP1-C_277 and Z_LMP2A-N_110 monospecific affibody controls. Although cisplatin’s antitumor efficacy was higher, mice treated with cisplatin showed significant weight loss compared to Z_LMP_110-277 and Z_LMP_277-110. Our results confirmed that Z_LMP_110-277 and Z_LMP_277-110, especially Z_LMP_277-110, significantly decreased NPC cell proliferation both in vitro and in vivo.

LMP1 and LMP2 are key EBV-encoded oncoproteins that activate numerous cell signaling pathways, such as ERK-MAPK, JNK/AP1, P13K, and NF-κB, influencing cell proliferation and other cellular responses [6,7]. Therefore, identifying the LMP1 and LMP2 signaling proteins that are involved in the underlying pathological mechanisms of EBV-associated NPC is essential for successful drug discovery and therapeutic targets. A potent small molecule inhibitor to disrupt LMP1 and LMP2 oncogenic pathways and the origin of tumor-initiating cells could potentially be a novel and independent prognostic modality and therapeutic target for patients with EBV-associated NPC. In the present study, Z_LMP_110-277 and Z_LMP_277-110 blocked the activation of the MEK/ERK/p90RSK downstream signaling pathway and inhibited nuclear translocation of c-Fos and c-Myc expression in NPC cell lines. Moreover, Z_LMP_110-277 and Z_LMP_277-110 were more potent in suppressing cellular proliferation as compared to Z_LMP1-C_277 and Z_LMP2A-N_110 in NPC cell lines. Our results further confirmed that bifunctional affibody molecules (Z_LMP_110-277 and Z_LMP_277-110) have the capacity to reduce cell proliferation and provide new indications for the therapeutic potential of Z_LMP_110-277 and Z_LMP_277-110 in EBV-associated NPC.

In summary, we constructed novel bifunctional affibody molecules (Z_LMP_110-277 and Z_LMP_277-110) and evaluated their binding affinity and specificity towards LMP1 and LMP2 using surface plasmon resonance, indirect immunofluorescence assays, co-immunoprecipitation assays, and near-infrared imaging both in vitro and in vivo. Moreover, in vitro and in vivo studies showed that Z_LMP_110-277 and Z_LMP_277-110, especially Z_LMP_277-110, showed significantly stronger inhibitory actions than Z_LMP1-C_277 and Z_LMP2A-N_110 in the proliferation of NPC cells. Moreover, Z_LMP_110-277 and Z_LMP_277-110 could inhibit phosphorylation of proteins modulated by the MEK/ERK/p90RSK signaling pathway, ultimately leading to suppression of oncogene nuclear translocations. Overall, our results showed that Z_LMP_110-277 and Z_LMP_277-110, especially Z_LMP_277-110, are promising novel prognostic indicators for molecular imaging and targeted treatment of EBV-associated NPC.

## 4. Materials and Methods

### 4.1. Materials

*Escherichia coli* (*E. coli*) strain BL21(DE3) and the *p*ET21a (+) vector were obtained from the American Type Culture Collection (ATCC), Novagen, and Amersham Pharmacia Biotech, respectively. Reagents included M13K07 helper phage (New England Biolabs, Ipswich, MA, USA), restriction endonucleases S*fI*, N*ot I*, N*deI,* and X*ho1*, and T4 DNA ligase (Thermo Fisher Scientific, Waltham, MA, USA), glutathione agarose, paraformaldehyde, and Triton X-100, isopropyl β-D-1-thiogalactopyranoside (IPTG) (Sigma Aldrich, Saint Louis, MO, USA), Ni-NTA agarose column (Qiagen, Valencia, CA, USA), and a bicinchoninic acid (BCA) protein assay kit (Beyotime, Beijing, China). Dulbecco’s Modified Eagle Medium (DMEM), Roswell Park Memorial Institute (RPMI) 1640, trypsin/ethylene diamine tetra-acetic acid (trypsin/EDTA), fetal bovine serum (FBS), and penicillin-streptomycin were purchased from Gibco. The TRIzol reagent and qPCR master mix were purchased from Takara Biomedical Technology (Beijing, China) Co., Ltd. Goat anti-mouse IgG conjugated to fluorescein isothiocyanate (FITC), Cy3 conjugate, and propidium iodide (PI) were obtained from Multi Sciences Biotech Co., Ltd. (Hangzhou, China). Dylight 755 (Thermo Fisher Scientific, Waltham, MA, USA), cell counting kit-8 (CCK-8) (Dojindo, Kumamoto, Japan), cell lysis buffer (Beyotime, Beijing, China), and nuclear and cytosolic extraction reagent (Applygen, Beijing, China) were obtained. Protease and phosphatase inhibitors were obtained from Roche. All antibodies (primary and secondary) for Western blotting were purchased from Cell Signalling Technology.

### 4.2. Design and Expression of Z_LMP_110-277 and Z_LMP_277-110

The Z_LMP2A-N_110 and Z_LMP1-C_277 affibody molecules were linked to obtain the Z_LMP_110-277 and Z_LMP_277-110 bifunctional affibody molecules, respectively. The two domains were spaced by a 15-amino-acid-long (Gly4Ser)_3_ linker region. The sequences of the genes encoding Z_LMP2A-N_110 and Z_LMP1-C_277 were isolated by PCR amplification and subcloned into plasmid *p*ET21a (+) using *NdeI* and *Xhol* sites. The DNA sequence of the bifunctional affibody molecules was determined by DNA sequencing.

After that, the gene sequences encoding Z_LMP_110-277 and Z_LMP_277-110 were inserted into expression plasmid *p*ET21a (+) encoding the His6 tag. The recombinant plasmids were transformed into the *E. coli* BL21 (DE3) strain, and the expressions of Z_LMP_110-277, Z_LMP_277-110, and SPA-Z scaffold (Z_WT_) were induced by 1 mM IPTG (Sigma Aldrich, Saint Louis, MO, USA). Media were cultured for 6 h, then harvested and re-suspended in PBS (pH 8.0). Cells were disrupted by sonication on ice and centrifuged at 12,000 rpm for 30 min at 4 °C. Then, the supernatants were filtered with a 0.45 μm syringe filter and subsequently loaded onto Ni NTA agarose columns (Qiagen, Valencia, CA, USA). Columns were washed with a gradient of washing buffers with increasing concentrations of imidazole (10, 50, 100, and 200 mmol/L). The bound proteins were eluted from the column with elution buffer (20 mM sodium phosphate, 0.5 M NaCl, 100 mM imidazole, pH 8.0). Subsequently, the proteins purity and molecular size were confirmed by sodium dodecyl sulfate-polyacrylamide gel electrophoresis (SDS-PAGE) and Western blotting using anti-His antibodies from Multi Sciences Biotech Co., Ltd. (Hangzhou, China).

### 4.3. Template-Based Protein Structure Prediction

The three-dimensional models of the fusion proteins were constructed using the SWISS-MODEL work space, and the biophysical properties were predicted by submitting the protein (amino acid) sequence of bifunctional affibody molecules on the ExPASY server (https://swissmodel.expasy.org/interactive, accessed on 15 November 2022).

### 4.4. Biosensor Analysis

A Biacore T200 (GE Healthcare, Uppsala, Sweden) was used to evaluate the binding affinity of bifunctional affibody molecules and their interactions with LMP1 and LMP2. In our laboratory, we have previously stored pET21a/EBV LMP1-C and *p*ET21a/EBV LMP2A-N, transformed into *E. coli* strain BL21-competent cells for soluble expression. The EBV LMP1-C and LMP2A-N were immobilized on the surface of the sensor chip CM5 (GE Healthcare) as ligands according to the method described previously [38]. After that, serial dilutions of samples were made, and the analytes were injected over the sensor chip surface to monitor protein-protein interaction. Binding curves were fitted to a 1:1 Langmuir model using BIAcore T200 evaluation 3.0.2 software.

### 4.5. Cell Culture

Two types of EBV-positive NPC cell lines, C666-1 and CNE-2Z (obtained from Taisheng Bio-Tech Guangzhou, China Co., Ltd., Guangzhou, China), and the EBV-negative HNE-2 (Shanghai Fu Life Industry Co., Ltd., Shanghai, China) were used. C666-1 and CNE-2Z express EBV proteins LMP1 and LMP2, and HNE-2 was used to investigate bifunctional affibody molecules binding specificity and antitumor effect. The cell lines (C666-1, CNE-2Z, and HNE-2) were cultured in DMEM and RPMI-1640 medium supplemented with 10% fetal bovine (FBS) and penicillin-streptomycin (100 units/mL and 0.1 mg/mL).

### 4.6. LMP1 and LMP2 Expression Level Analysis in NPC Cells

All cells were cultured, and total ribonucleic acid (RNA) was extracted from C666-1, CNE-2Z, and HNE-2Z cells using the TRIzol reagent Takara from Biomedical Technology Co., Ltd. (Beijing, China). RNA is reverse transcribed into complementary deoxyribonucleic acid (cDNA), followed by two sets of primers specifically designed for each target gene and the Power SYBR Green PCR Master Mix (Thermo Fisher Scientific, Waltham, MA, USA). The results were analyzed using QuantStudio Real-Time PCR software (Life Technologies, Carlsbad, CA, USA).

Furthermore, Western blotting was performed to confirm LMP1 and LMP2 expression levels in NPC cells. Briefly, cells were cultured and harvested, washed with ice-cold PBS, and lysed directly in lysis buffer. After that, equal amounts of proteins are separated by SDS-PAGE on 15% gels and transferred to polyvinylidene difluoride (PVDF) membranes (Millipore, Burlington, MA, USA). Membranes were blocked in 5% skim milk in PBS with 0.5% Tween 20 (PBST) for 2 h, incubated with primary antibodies, anti-LMP1 (Abcam 136633) or rabbit anti-LMP2A-NCD prepared in-house, overnight at 4 °C, and detected with a fluorescently tagged secondary antibody. Protein bands were visualized using the Western Blotting Imaging System (Clinx, Shanghai, China), and protein expression was quantified by ImageJ 1.33 software (National Institutes of Health). Glyceraldehyde 3-phosphate dehydrogenase (GAPDH) was used as an internal control.

### 4.7. Indirect Immunofluorescence Assay

An immunofluorescence assay was used to analyze the binding specificity of bifunctional affibody molecules to NPC cells. CNE-2Z, C666-1, and HNE-2 cells were grown on cover slips using cell culture dishes (1 × 10^5^ cells/well) for 24 h. Then, cells were treated with 100 μg/mL of affibody molecules (Z_LMP_110-277, Z_LMP_277-110, Z_LMP1-C_277, and Z_LMP2A-N_110) or Z_WT_ affibody (negative control) and incubated for 3 h at 37 °C. After washing with PBS, cells were fixed with 4% paraformaldehyde and permeabilized using 0.3% Triton X-100 at room temperature. The cells on cover slips were then blocked with blocking buffer for 1 h, followed by incubation with the primary antibody (mouse anti-His-tag mAb) overnight at 4 °C. Afterwards, cells are stained with FITC-conjugated goat anti-mouse IgG secondary antibodies at 37 °C for 1 h. The cell nuclei were stained with propidium iodide (PI) dye, and cells were imaged using a confocal fluorescence microscope (Nikon C1-i, Tokyo, Japan).

To further prove the specificity of bifunctional affibody molecules to LMP1 and LMP2 proteins, colocalization analyses using double immunofluorescence were performed. The procedure is similar to that given above.

### 4.8. Immunoprecipitation

Briefly, C666-1 cells received treatment with bifunctional or monospecific affibody molecules (100 μg/mL) and were incubated at 37 °C for 3 h. Cells were washed in ice-cold PBS and then lysed with RIPA buffer containing protease inhibitors. Then, anti-LMP1 (Abcam 136633) or anti-LMP2 (Abcam, Clone15F9) antibodies were combined with disuccinimidyl suberate bound to protein A/G plus agarose. Afterwards, the pellets were resuspended in SDS sample buffer and analyzed by Western blotting.

### 4.9. In Vivo Tumor Imaging

Near-infrared (NIR) imaging was used to investigate the distribution and tumor-specific targeting ability of bifunctional affibody molecules in nude mice. C666-1, CNE-2Z, and HNE-2 cells (1 × 10^7^ cells/100 μL PBS, respectively) were subcutaneously injected into the right forearm of nude mice (n = 5 per group). The tumor-bearing mice were used for imaging when tumor volume reached 300–500 mm^3^ (around 3–4 weeks after injection). Z_LMP_110-277, Z_LMP_277-110, Z_LMP1-C_277, Z_LMP2A-N_110, and SPA-Z scaffold (Z_WT_) were labeled with Dylight-755 (Thermo Fisher Scientific, Waltham, MA, USA) according to the manufacturer protocol. For in vivo imaging, 100 μg of Dylight-755-labeled affibody molecules dissolved in PBS (150 μL) were administered through the tail vein and imaged at different time points after injection using the NIR imaging system (Cri Maestro 2.10, Massachusetts, USA). The tumor/skin tissue signal intensity ratios were analyzed at various time points post-injection.

### 4.10. In Vitro Efficacy of Z_LMP_110-277 and Z_LMP_277-110

Cell viability assays were performed to evaluate the efficacy of Z_LMP_110-277 and Z_LMP_277-110 in EBV-positive NPC cells (C666-1 and CNE-2Z). C666-1 and CNE-2Z cells (1x10^4^ cells/well) were cultured in 96-well plates, followed by treatment with Z_LMP_110-277, Z_LMP_277-110, Z_LMP1-C_277, or Z_LMP2A-N_110 at increasing concentrations (0.6, 1.2, 2.5, 5, 10, and 20 μM). The anticancer drug cisplatin and the SPA-Z scaffold (Z_WT_) were used as positive and negative controls, respectively. Cell viability was determined after each incubation period (12, 24, 36, 48, and 72 h). Next, 10 µL of cell counting kit-8 (CCK-8) solution per well (Dojindo, Kumamoto, Japan) was added and subjected to an additional 30 min of incubation. The absorbance at 450 nm was determined using a microplate reader (Synergy HT; BioTek, Winooski, VT, USA). The half maximal inhibitory concentration (IC50) values were calculated by Graph Pad Prism software (Graph Pad Software, Inc., San Diego, CA, USA).

### 4.11. Colony Formation Assay

The cell proliferation of NPC cells was evaluated using a colony formation assay. Briefly, C666-1, CNE-2Z, and HNE-2 cells were seeded at (1 × 10^3^ cells/well) in 6-well plates and then treated with Z_LMP_110-277, Z_LMP_277-110, Z_LMP1-C_277, or Z_LMP2A-N_110 (100 μg/mL) for 14 days. Subsequently, cells were washed with PBS, fixed with 4% paraformaldehyde, and stained with 0.1% Crystal Violet Staining Solution (Amresco, Solon, OH, USA). A light microscope was used to count the colonies formed (a group of more than 50).

### 4.12. Flow Cytometry Assay

Cell cycle alteration was detected using flow cytometry. C666-1, CNE-2Z, and HNE-2 cells were harvested after treatment for 24 h with Z_LMP_110-277, Z_LMP_277-110, Z_LMP1-C_277, or Z_LMP2A-N_110 (100 μg/mL) and then fixed in 70% ethanol overnight at 4 °C. Fixed cells were washed in PBS and stained with propidium iodide (PI) (50 μg/mL PI, 100 μg/mL Rnase) for 30 min. Distributions of the cell cycle were determined using a FACS Calibur flow cytometer (BD, Biosciences, San Jose, CA, USA), and the resulting data were analyzed using ModFit LT version 3.0 software.

### 4.13. Western Blotting Assay for Cell Signaling Pathway Proteins

C666-1 and CNE-2Z cells were seeded at (1 × 10^5^ cells/well) in 6-well plates and incubated for 36 h with Z_LMP_110-277, Z_LMP_277-110, Z_LMP1-C_277, Z_LMP2A-N_110, or SPA-Z scaffold (Z_WT_) (10 µM). After that, the cells were detached and lysed in lysis buffer (RIPA buffer, Beyotime, Beijing, China) containing protease and phosphatase inhibitors. Then, proteins were separated by 15% SDS-PAGE and transferred to a PVDF membrane. The membranes were briefly blocked with 5% skim milk and incubated with a primary antibody (Appendix A) at 4 °C overnight. After washing, the membranes were incubated with secondary antibodies and then imaged using the Western blotting imaging system. GAPDH served as a reference gene.

### 4.14. In Vivo Antitumor Efficacy

C666-1, CNE-2Z, and HNE-2 tumor-bearing mice were divided randomly into 5 groups (n = 5 per group) and treated by tail vein injection every three days for 30 days with the following 0.1 mL: (I) PBS; (II) SPA-Z scaffold (Z_WT_) 100 nmol/kg; (III) Cisplatin 5 mg/kg (used as a positive control); (IV) Z_LMP_110-277 100 nmol/kg; (V) Z_LMP_277-110 100 nmol/kg; (VI) Z_LMP1-C_277 100 nmol/kg; and (VII) Z_LMP2A-N_110 100 nmol/kg. The therapeutic efficacy of the different regimens was monitored by daily measurements of tumor sizes and body weight. At day 30, all mice were sacrificed, and the tumors were surgically removed carefully. The tumors and major organs were collected and frozen at −80 °C for further use.

### 4.15. Statistical Analysis

The data were expressed as mean ± standard deviation (SD). Statistical analysis of the results was performed with a Student’s t-test, and a probability (*p*) value < 0.05 was considered statistically significant. All the graphs were performed using GraphPad Prism software.

## Figures and Tables

**Figure 1 ijms-24-10126-f001:**
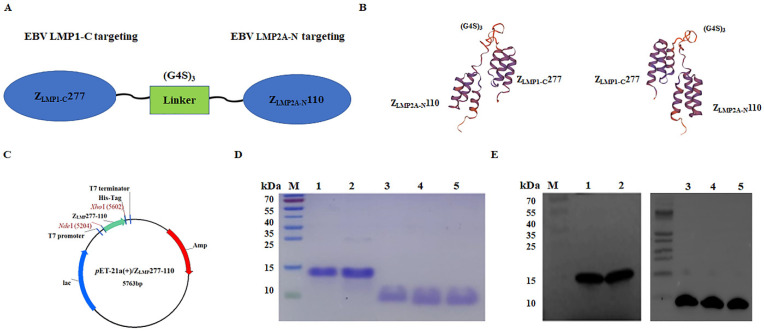
Expression and purification of bifunctional affibody molecules. (**A**) A schematic diagram showing two monomers linked by (G4S)_3_ to construct bifunctional affibody molecules. (**B**) Predicted 3D models of bifunctional affibody molecules. (**C**) Schematic diagram of the *p*ET21a(+)/bifunctional affibody recombinant plasmid. (**D**) The purified bifunctional and monospecific affibody molecules were analyzed by SDS-PAGE. (**E**) The purified proteins were confirmed by Western blotting analysis. M, protein ladder; 1, Z_LMP_110-277; 2, Z_LMP_277-110; 3, Z_LMP1-C_277; 4, Z_LMP2A-N_110; and 5, SPA-Z scaffold (Z_WT_). Experiments were performed in triplicate.

**Figure 2 ijms-24-10126-f002:**
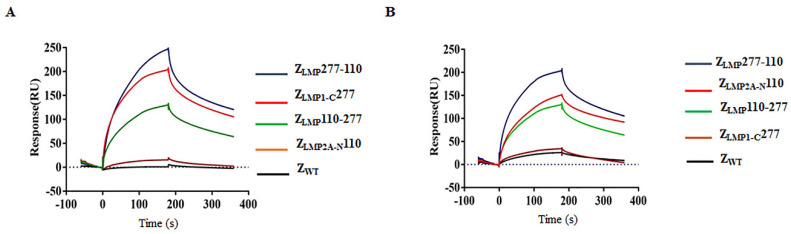
Binding of bifunctional affibody molecules to target proteins. (**A**,**B**) Representative sensograms measuring the binding affinity of Z_LMP_110-277, Z_LMP_277-110, Z_LMP1-C_277, and Z_LMP2A-N_110 towards LMP1 and LMP2, respectively. The SPA-Z scaffold (Z_WT_) affibody was used as a negative control. Experiments were performed in triplicate.

**Figure 3 ijms-24-10126-f003:**
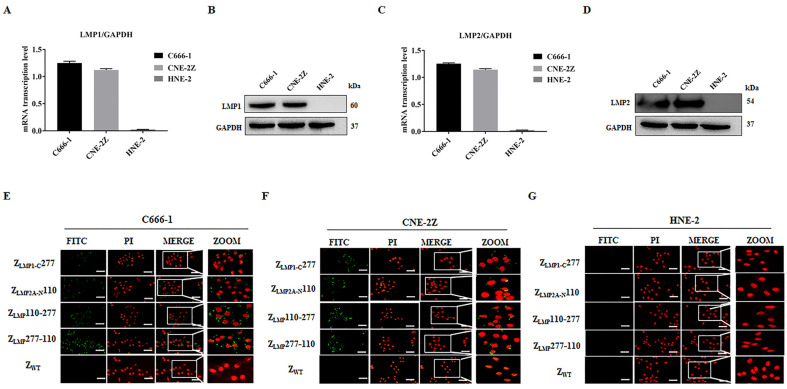
Analysis of the binding specificity of bifunctional affibody molecules to LMP1 and LMP2. (**A**,**C**) A qPCR analysis of LMP1/GAPDH and LMP2/GADPH in C666-1, CNE-2Z, and HNE-2 cell lines. (**B**,**D**) Western blotting protein expression analysis of LMP1 and LMP2 in C666-1, CNE-2Z, and HNE-2 cell lines. The data are given as the mean ± SD (n = 3). (**E**,**F**) NPC-positive cell lines were incubated with bifunctional or monospecific affibody molecules. Corresponding antibodies are labeled with FITC (green), and the nuclei of cells are stained with PI (red) (400×). (**G**) HNE-2 was set as the control cell line (NPC-negative), and SPA-Z scaffold (Z_WT_) affibody was used as the negative control. Experiments were performed in triplicate.

**Figure 4 ijms-24-10126-f004:**
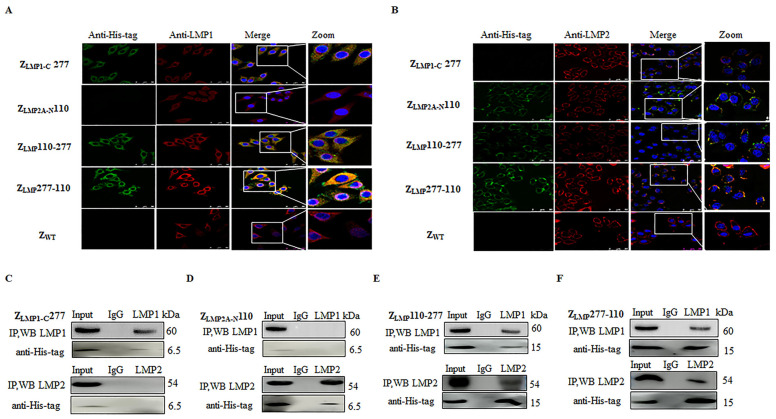
Bifunctional affibody molecules interacted with both LMP1 and LMP2 proteins. (**A**,**B**) C666-1 cells were incubated with bifunctional or monospecific affibody molecules for three hours and analyzed by confocal microscopy. The rabbit anti-LMP1 or anti-LMP2 mAb and the mouse anti-His-tag mAb were used as primary antibodies. The goat anti-rabbit antibody conjugated with Cy3 (red) and the goat anti-mouse antibody conjugated with FITC (green) were used as secondary antibodies. The cell nuclei were stained by Hoechst3342 (blue) (400×). The merge images show the co-localization of bifunctional and monospecific affibody molecules with LMP1 and LMP2 (yellow). (**E**,**F**) Z_LMP_110-277 and Z_LMP_277-110 interact with both LMP1 and LMP2, and then an IP was performed with anti-LMP1 and anti-LMP2 antibodies. (**C**,**D**) Z_LMP1-C_277 and Z_LMP2A-N_110 complexed with LMP1 or LMP2 following an IP. Western blotting analysis was performed and incubated with rabbit anti-LMP1, anti-LMP2, or anti-His-tag mAb. IgG served as a negative control. Experiments were performed in triplicate.

**Figure 5 ijms-24-10126-f005:**
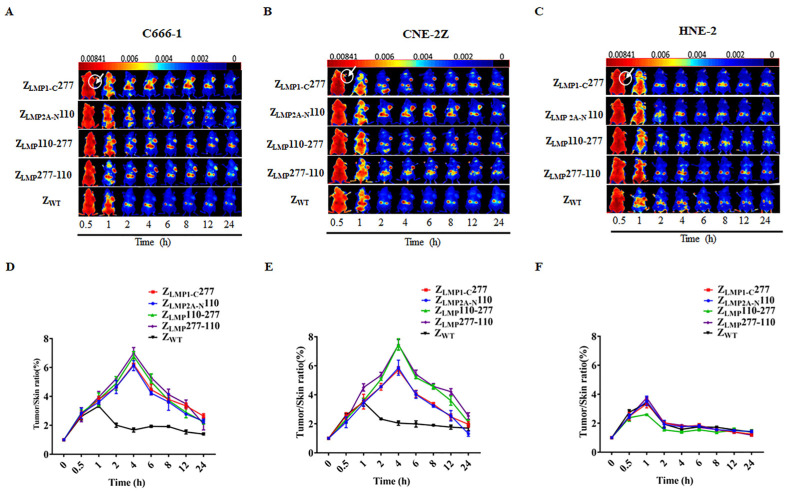
Tumor uptake of bifunctional affibody molecules by NPC-bearing nude mice. Tumor-bearing nude mice were generated with cell lines C666-1 (**A**), CNE-2Z (**B**), and HNE-2 (**C**). NIR imaging was performed at different time points post-injection with Dylight-755-labeled bifunctional and monospecific affibody molecules. Dylight-755-labeled SPA-Z scaffold (Z_WT_) affibody molecules were used as a negative control. (**D**–**F**) Tumor/skin ratios were calculated at various time points post-injection of the indicated agents in C666-1, CNE-2Z, and HNE-2 tumor-bearing mice. Data are displayed as the mean ± SD (n = 3).

**Figure 6 ijms-24-10126-f006:**
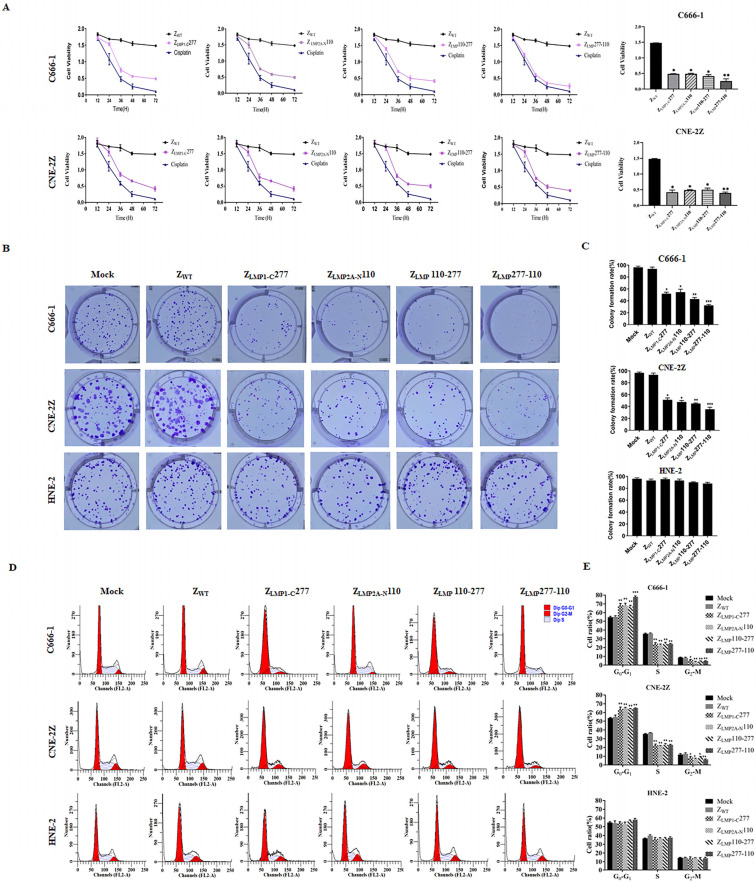
The efficacy of bifunctional affibody molecules in vitro. (**A**) Cell viability assays were performed to determine the viability of C666-1 and CNE-2Z for the time periods indicated, with bar graphs showing cell viability 72 h after treatment. SPA-Z scaffold (Z_WT_) was set as the affibody negative control. * *p* < 0.05; ** *p* < 0.01 compared to Z_WT_. (**B**) Colony formation assays were performed to study the long-term effects of bifunctional and monospecific affibody molecules on NPC cell proliferation. (**C**) Statistical analysis of the colony formation assay. * *p* < 0.05; ** *p* < 0.01; *** *p* < 0.001 compared to Z_WT_. (**D**) C666-1, CNE-2Z, and HNE-2 cells were treated with 100 μg/mL affibody or Z_WT_ (negative control), and the cell cycle was analyzed with flow cytometry and PI staining. (**E**) Percentages of cells in the G0/G1, S, and G2/M phases were calculated from three independent experiments. * *p* < 0.05; ** *p* < 0.01; *** *p* < 0.001 compared to SPA-Z scaffold (Z_WT_). Experiments were performed in triplicate.

**Figure 7 ijms-24-10126-f007:**
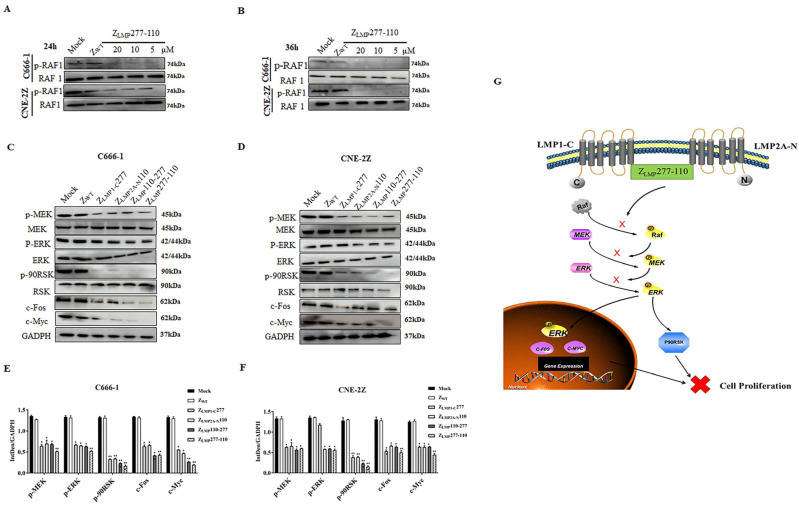
Bispecific affibody molecules inhibit proliferation by the MEK/ERK/p90RSK pathway in NPC cells. (**A**,**B**) p-RAF was downregulated in a concentration- and time-dependent manner after treatment with Z_LMP_277-110 in both C666-1 and CNE-2Z. (**C**,**D**) Bifunctional and monospecific affibody molecules downregulated MEK/ERK/p90RSK signaling proteins and transcription factors. (**E**,**F**) Quantification of Western blotting by ImageJ. * *p* < 0.05; ** *p* < 0.01; compared to SPA-Z scaffold (Z_WT_). (**G**) A schematic illustration of the bifunctional affibody molecule blocking the MEK/ERK/p90RSK signal pathway. Experiments were performed in triplicate.

**Figure 8 ijms-24-10126-f008:**
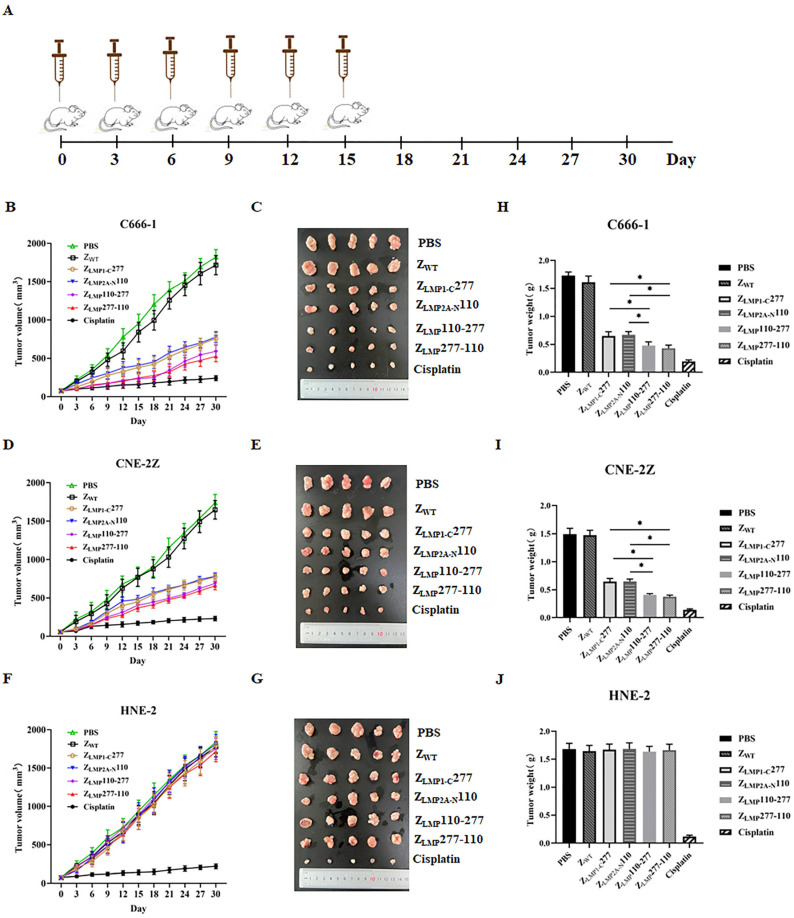
Therapeutic efficacy of Z_LMP_110-277 and Z_LMP_277-110 in NPC-bearing nude mice. (**A**) Schematic illustration of the nude mouse treatment. (**B**,**C**) Tumor-bearing mice were divided randomly into five groups and then administered different treatments (PBS, SPA-Z scaffold (Z_WT_), Z_LMP_110-277, Z_LMP_277-110, Z_LMP1-C_277, Z_LMP2A-N_110, and cisplatin). (**D**,**E**) The photos of tumors separated from NPC-positive cell lines (C666-1 and CNE-2Z). (**F**) Tumor volumes and (**G**) tumors removed from mice in HNE-2 cells. Tumors from cell lines (**H**) C666-1, (**I**) CNE-2Z, and (**J**) HNE-2 were compared. The data are given as mean ± SD (n = 5). * *p* < 0.05 compared to control.

**Table 1 ijms-24-10126-t001:** Kinetic data from the SPR binding affinity analysis of the affibody Molecules to LMP1.

Affibody	Ka (1/Ms)	Kd (1/s)	KD (M)
Z_LMP_110-277	124.1	4.070 × 10^−3^	3.28 × 10^−5^
Z_LMP_277-110	2600	2.124 × 10^−3^	8.08 × 10^−7^
Z_LMP1-C_277	841.8	3.066 × 10^−3^	3.56 × 10^−6^
Z_LMP2A-N_110	8.659	1.723 × 10^−3^	1.96 × 10^−4^
Z_WT_	4.41 × 10^−4^	2.464 × 10^−3^	1.35 × 10^−1^

**Table 2 ijms-24-10126-t002:** Kinetic data from the SPR binding affinity analysis of the affibody Molecules to LMP2.

Affibody	Ka (1/Ms)	Kd (1/s)	KD (M)
Z_LMP_110-277	480.8	5.029 × 10^−3^	1.04 × 10^−5^
Z_LMP_277-110	987.5	1.929 × 10^−3^	1.92 × 10^−6^
Z_LMP1-C_277	3.96 × 10^−2^	2.464 × 10^−3^	6.47 × 10^−4^
Z_LMP2A-N_110	867	3.419 × 10^−3^	3.92 × 10^−6^
Z_WT_	4.47 × 10^−4^	5.971 × 10^−5^	1.33 × 10^−1^

Abbreviations: Ka, Association rate constant; Kd, Dissociation rate constant; KD, Dissociation equilibrium constant.

## Data Availability

The original contributions presented in the study are included in the article/Appendix A. Further inquiries can be directed to the corresponding author.

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
