# Peer review of "Novel Bifunctional Affibody Molecules with Specific Binding to Both EBV LMP1 and LMP2 for Targeted Therapy of Nasopharyngeal Carcinoma"

_ijms, 2023, doi:10.3390/ijms241210126_

Round 1

Reviewer 1 Report

The authors demonstrated that they developed a bispecific affibody which is specific to LMP1 and LMP2. Their data shows specificity of the affibody and further demonstrates the efficacy of the bispecific in comparison to the monospecific affibodies. The authors have previously published three other articles on the same topic where they demonstrate the development and effectiveness of the monospecific affibodies. However, in this communication their work focuses on the bispecific affibodies and their slight advantage over the monospecific affibodies through in vitro and in vivo studies. The writing and organization of the article is satisfactory; however, some improvements are required in the demonstration of figures and description in figure legends, as listed below.

1.       In Figure 6A, the authors display an array of cell viability graphs, which is great but as each treatment group is displayed on a separate graph, it is hard to perform a direct compare between the groups, as claimed in writing in the result section. I suggest that they include a bar graph with all the treatment groups for easy visualization of the data. They can pick one point (72 hr) and add these graphs with statistical comparison to the figure.

2.       In figure 6E the bars for ZWT are in white with no outline, this makes it difficult to see. I suggest adding a pattern and outline to the bars to make the visualization easy on the eye.

3.       Fig 7E/F are labeled as quantitative analysis of the western blots in 7C/D, there need to be a description of how this analysis was performed in the figure legend.

4.       In Figure 8G/H, the statistical comparison is performed between Cisplatin groups and others, as this article focuses on how the bispecific improve the anti-tumor effect in comparison to the mono-specifics. I believe the statistical comparison should be made between the groups of mono vs bi – specific affibodies to make the data more meaningful.

5.       I also suggest adding a line schematic of tumor study in figure 8.

6.       Did the authors perform any in vivo tumor studies where they followed the tumors after thirty days with continued treatment to see if the tumors keep growing or cease to exist after prolonged treatment.

7.       In materials and methods sub-section 4.12, the authors need to include the information about the treatment time for the flow cytometry data.

8.       In general, the authors need to improve the figure legends to include information about the treatment times/doses and other important information that relates to the set up of the experiment. The authors also don’t mention how many times each experiment was repeated and if the data presented is representative data from one experiment or pooled data in case the experiment was repeated multiple times.

Author Response

Thank you for giving us the opportunity to submit a revised draft of our manuscript. We appreciate the time and effort that the reviewer dedicated to providing feedback on our manuscript and we are grateful for the insightful comments on and valuable improvements to our paper.

Reviewer 2 Report

EBV constitutes a significant global health burden, and the absence of approved prophylactic vaccines or therapeutic agents against EBV further exacerbates the situation. This study holds great significance as its objective is to develop and characterize bispecific affibodies capable of binding to both LMP1 and LMP2 latent antigens, thus exhibiting the potential to inhibit virus replication and tumor growth. The findings presented in this manuscript build upon the author's previous report on affibodies LMP2A-N and LMP1-C, wherein they have introduced a linker to create a bispecific affibody capable of binding to both LMP1 and LMP2. Comprehensive in vitro and in vivo characterizations have been conducted.

While the authors have effectively presented their data and written the manuscript, certain critiques should be addressed:

1.

In addition to examining the inhibition of cell/tumor growth through the binding of LMP1 and LMP2 specific affibodies, it would be beneficial for the authors to demonstrate the effect on virus replication both in vitro and in vivo, employing methods such as qPCR.

2.

It is important to address the variability in the intensity of anti-LMP1 or anti-LMP-2 (red) staining across the different panels in Fig 4A and B.

3.

Line 205: The mention of "100ug" in the tumor xenografts development procedure contrasts with the method section, which specifies "100umol." Please provide clarification and ensure consistency.

4.

Line 268: To justify the claim that downregulation of the signal transduction pathway inhibits EBV replication, it is essential to provide data and incorporate appropriate experiments as direct evidence.

Author Response

(The authors gave the same response as above.)

Reviewer 3 Report

In their manuscript, Kamara et al. describe the construction an use of bifunctional affibodies against LMP1 and LMP2 on nasopharyngal carcinomas. In principal, this appears as a very promising approach. However, the effects of the bifunctional compared to the monofuctional reagents are comparatively small. This needs to be discussed. In addition, the manuscript is extremely difficult to read and understand. The general style as well as the English needs extensive editing. In detail:

1.      English

2.      The experiments appear to be carried out only once. To reproduce results is an essential part of GLP. Here one could argue that two different tumor lines were tested. Nevertheless, this is a very negative point of the manuscript.

3.      Abstract: several abbreviations are used and should be spelt out when used the first time. The same is true for the rest of the text.

4.      Line 61: anti-HER2 and anti-EGFR1 is meant.

5.      Line 69: what does targeting in this context mean? Probable consisting of or the like is meant. Similar mistakes are found throughout the text.

6.      Affibodies should be shortly introduced. Also how they were selected and characterized. Referring to another publication is not helpful here.

7.      Line 108: the His tag should be mentioned. Its use comes to a surprise later in the results.

8.      Figure 1 D and E. In both cases isolated proteins are displayed but the nomination is different.

9.      ZWT appears suddenly and is not explained. Also not in M&M.

10.   Line 134: SPR should be spelled out when used the first time.

11.   Line 140: LMP1 and LMP2 suddenly appear. It should be mentioned that purified recombinant proteins are used.

12.   Line 171: the authors claim that they investigate “intracellular” interaction. To my understanding and if indeed functional the affibodies should bind to the molecule outside of the cells and not intracellular.

13.   Figure 3: the quality of the fluorescence picture is extremely low. In some, nothing can be recognized. The same is true for Figure 4.

14.   The paragraph on co-immune precipitation is completely ununderstandable and should be rewritten.

15.   Line 207: NIR should be spelled out.

16.   Figure 5 D-F is too small.

17.   Suppl. Fig. S3: controls with HNE and ZWT are missing.

18.   Firgure 6 A, C and D is too small.

19.   Figure 7: HNE control is missing. Panels E,F and G are too small.

20.   Line 302: the authors claim no toxicity is observed. How was that determined. I assume it was judged on the general appearance of the mice. This is not sufficient for this claim.

21.   Figure 8: the experiments appear not to be reproduced.

22.   Line 373: what is CCK-8 indicating.

23.   Line 510: why were the proteins crosslinked?

24.   Fig. S2: the panels are much too small.

25.   Fig. S3: the figure is much too small. Nothing can be read.

see above

Author Response

(The authors gave the same response as above.)

Round 2

Reviewer 3 Report

In their revised version, the authors have largely ignored my comments. They provide a very wordy rebuttal but do not argue to the point or add a correction where necessary. They explain some terms, which are obvious but ignore the reason why this point is made. They hardly ever change the text. Here are some of the major points: 1. the English was not edited. Some of the sentences state complete nonsense. Serious corrections should be made. 2. The experiments are not reproduced. The arguments the authors provide are not valid. If the experiments were carried out more than once it should be stated in the text or the legend. In addition, the number of fields of view should be given for the microscopy pictures. 3. The authors claim that the affibodies recognize intracellular epitopes of LMP1 and LMP2. This is confusing. How should an affibody recognize an intracellular antigen when simply given to the outside of the cell without membrane permeabilization. How should that work in vivo.

Thank you for giving us the opportunity to submit a revised draft of our manuscript.
We appreciate the time and effort that the reviewer dedicated to providing feedback
on our manuscript and we are grateful for the insightful comments on and valuable
improvements to our paper. We have incorporated some of the suggestions made by
the reviewer. Those changes are highlighted within the manuscript. Please see below,
in blue, for a point-by-point response to the reviewer comments and concerns.
1. English
We appreciate the time and effort that the reviewer has dedicated to providing
valuable feedback on our manuscript. We apologize for the poor writings; however,
we asked a native English speaker to help polish our article. The manuscript has
been revised extensively according to the reviewers' constructive suggestions.

See above
2. The experiments appear to be carried out only once. To reproduce results is an
essential part of GLP. Here one could argue that two different tumor lines were tested.
Nevertheless, this is a very negative point of the manuscript.
(1) We appreciate the reviewer comments, however, the experiments were conducted
more than once to make sure our results are correct and not a fluke or wrongly
measure.
(2) Our laboratory has been engaged in the research of affibody molecules for more
than a decade, and has obtained series of affibody molecules with anti-tumor effects
using phage display technology, such as HPV16E7, EBV LMP2 [1-3]. Additionally,
HER2-specific affibody molecules labeled with 111In have shown to be useful for
imaging of HER2 expression in breast cancer and therapeutic applications [4], and
also the first candidate of therapeutic affibody (ABY-035) was tested in clinical trials
and proven to be safe and well tolerated in healthy volunteers (www.Affibody.se).
The binding affibody molecules were obtained by the same technology and method,

and in this study, our results demonstrates that the affibody molecules were
investigated for their binding specificity towards LMP1 and LMP2 in vitro and in
vivo and does have antitumor effects on NPC-positive cell lines.
(3) We used EBV DNA-positive nasopharyngeal cancer cell lines C666-1 and CNE-
2Z and EBV DNA- negative cell HNE-2 as research cells, and performed qPCR and
Western Blot to test and verify the Epstein-Barr virus genes and expressed proteins
carried by the cell lines respectively (Fig. 3ABCD).

It is not necessary to summarize the experiments. The still were not reproduced. This is especially valid for the in vivo experiments.
3. Abstract: several abbreviations are used and should be spelt out when used the
first time. The same is true for the rest of the text.
Thanks for your comment. We have made changes in the abstract section accordingly
in the revised manuscript.

CCK8 ist still not spelled out.
4. Line 61: anti-HER2 and anti-EGFR1 is meant.
This is a research report on the ability of both anti-HER2 and anti-EGFR1 small
molecule that effectively inhibit the proliferation of metastatic breast cancer cells.
The results of the study found that bifunctional molecule have more pronounce effect,
which provides a reference for the design and development of this study [5].

The mistake is now at line 63. The authors should have noticed this mistake by themselves. anti-HER2 and anti-EGFR1 is meant.
5. Line 69: what does targeting in this context mean? Probable consisting of or the

like is meant. Similar mistakes are found throughout the text.
Thanks for your comment. Targeted therapy is a type of cancer treatment that targets
proteins that control how cancer cells grow, divide, and spread. It is the foundation
of precision medicine. In this study, the target proteins of the affibodies selected in
this manuscript are EBV LMP1 and LMP2. It has been confirmed that both C666-1
and CNE-2Z of nasopharyngeal cancer cells express EBV LMP1 and LMP2 proteins,
so they are used as targets for affibody therapy research.

I do not need any explanation what targeting means. The authors claim here that they target a single chain variable fragment or CDR3 region, which is complete nonsense. They should have noticed this mistake themselves and correct it instead of explaining something obvious.

6. Affibodies should be shortly introduced. Also how they were selected and
characterized. Referring to another publication is not helpful here.
Thanks for your comment. We have made changes accordingly in the revised
manuscript.
(1) Affibody molecules are small (6.5-kDa) affinity proteins based on a three-helix
bundle domain framework. Since their introduction 20 years ago as an alternative to
antibodies for biotechnological applications.
(2) A random affibody library was prepared [6] by PCR amplifcation from a wild
SPA-Z scaffold template by using the random primers encoding helices 1 and 2 of
the Z domain. Based on the degree of the amino acid sequence and structure of wild-
type SPA-Z scaffold (ZWT), primers were designed base on the coding sequences
corresponding to its three helical regions. The SPA coding sequence that could affect
amino acid changes was amplified using polymerase chain reaction (PCR) and is
termed SPA-N. The coding sequence of SPA-N was cloned into phagemid
(pCANTAB5E) using the SfI and NotI sites to construct the phagemid vector
pCANTAB5E/ SPA-N and transformed into competent cell Escherichia coli TG1
(DE3). The naive affibody molecule library cloned into the vector was found to
contain approximately 1× 108 and with 100% SPA-Z scaffold. After evaluating the
capacity and randomness of the inserted affibody library, the phage stocks were used
to pan potential affibody that selectively binds to LMP1 and LMP2 terminal domain
using phage display technology.

Part of this response should have been added to the text. In the results, it should be mentioned that the present affibodies are based on a SPA scaffold and selected by phage display. The more technical aspects mentioned her should be added to the M&M section.
7. Line 108: the His tag should be mentioned. Its use comes to a surprise later in

the results.
Thanks for your valuable comments. We have made changes accordingly in the
revised manuscript (Line 108).

8. Figure 1 D and E. In both cases isolated proteins are displayed but the nomination
is different.
In Fig 1D, the His-tagged proteins expressed were purified successfully by affinity
chromatography using Ni-NTA resin and confirmed by SDS-PAGE analysis. In Fig
1E, Western blotting further confirmed that fusion proteins specifically recognized
by the anti-His-tag mouse monoclonal antibody.
Again a complete unnecessary explanation. In D: pET211a(+)/ZLMP110-277 ……………. is indicated. This is certainly wrong because purified proteins without plasmids are displayed. The plasmid name should be deleted. E: is correct. The authors should have noticed this by themselves instead of writing a long answer.

9. ZWT appears suddenly and is not explained. Also not in M&M.
Thanks for your valuable comments. We have made changes accordingly in the
revised manuscript.
When Zwt is used first time it should be mentioned that it consist of the SPA scaffold.

10. Line 134: SPR should be spelled out when used the first time.
Thanks for your valuable comments. We have made changes accordingly in the
revised manuscript as “surface plasmon resonance (SPR)” Line 134.
11. Line 140: LMP1 and LMP2 suddenly appear. It should be mentioned that
purified recombinant proteins are used.
Thanks for your comments. We have made changes accordingly in the revised
manuscript as “purified recombinant proteins” Line 140.
Instead of indicating that LMP1 and LMP2 was produced as recombinant proteins, the authors now indicate that the affibodies are recombinant purified proteins. This is obvious but from where is LMP1 and LMP2 derived from. What a mess.

12. Line 171: the authors claim that they investigate “intracellular” interaction. To
my understanding and if indeed functional the affibodies should bind to the molecule
outside of the cells and not intracellular.
Thanks for your comments. The latent membrane proteins of EBV are
transmembrane proteins. In this study, the cytoplasmic domains of LMP1 and LMP2
were selected as the target antigens for screening affibodies. Therefore, affibody
molecules targets the cytoplasmic domains that bind LMP1 and LMP2 in the
cytoplasm. Confocal immunofluorescence and co-immunoprecipitation assays were

performed to confirm the in vitro intracellular interaction of bifunctional affibody
molecules to LMP1 and LMP2 proteins.
See above.

13. Figure 3: the quality of the fluorescence picture is extremely low. In some,
nothing can be recognized. The same is true for Figure 4.
Thanks for your valuable comments. We have made changes in figure 3 and 4
accordingly in the revised manuscript.
The authors claim that one construct gives stronger binding. This is not obvious and should be shown by quantitation using photon counting.

14. The paragraph on co-immune precipitation is completely ununderstandable and
should be rewritten.
Thanks for your valuable comments. We have made changes accordingly in the
revised manuscript.
15. Line 207: NIR should be spelled out.
Thanks for your comments. We have made changes accordingly in the revised
manuscript
16. Figure 5 D-F is too small.
Thanks for your comments. We have made changes accordingly in the revised
manuscript.
17. Suppl. Fig. S3: controls with HNE and ZWT are missing.
The bifunctional affibody molecules showed no obvious inhibitory effect on HNE
cells, whereas ZWT had no effect on any of the three cell lines used in this study.
Therefore, the results were not shown.
This should be mentioned in the text.

18. Firgure 6 A, C and D is too small.
Thanks for your comments. We have made changes accordingly in the revised
manuscript.

19. Figure 7: HNE control is missing. Panels E, F and G are too small.
(1) Thanks for your comments. In this study, our research purpose is focus on the
protein inhibition of the signaling pathways activated by LMP1 and LMP2 and the
proliferation and phenotypic changes in nasopharyngeal carcinoma positive cell
lines (C666-1 and CNE-2Z).
(2) Data from previously published study from our lab [7], showed that monospefic
affibody (ZLMP1-C277) treatment did not decrease phospho-Raf-1(Ser338) in a
concentration- and time-dependent manner in HNE-2 cell line. Western blotting
showed that ZLMP1-C277 did not induced a reduction of phospho-MEK1/2
(Ser217/Ser221), phospho-ERK1/2(Thr202/Thr204), phospho-p90RSK(Ser380)
and transcription factor c-Fos levels in HNE-2 cell line (result are found in
supplementary file in previous published article).
(3) We have made changes accordingly in the revised manuscript to panels E, F and
G
20. Line 302: the authors claim no toxicity is observed. How was that determined.
I assume it was judged on the general appearance of the mice. This is not sufficient
for this claim.
We apologize for the inconvenience caused and the manuscript has been revised
accordingly to clarify.
21. Figure 8: the experiments appear not to be reproduced.
Thanks for your comments. Our laboratory has been engaged in the research of
affibody molecules for more than a decade, and has obtained series of affibody
molecules with anti-tumor effects using phage display technology, such as HPV16E7,
EBV LMP2 [1-3]. In this study, the binding affibody molecules were obtained by

the same technology and method, and our results demonstrates that the affibody
molecules were investigated for their binding specificity towards LMP1 and LMP2
in vitro and in vivo and does have antitumor effects on NPC-positive cell lines.
See aabove

22. Line 373: what is CCK-8 indicating.
Thanks for your comments. Cell Counting Kit-8 (CCK8) was used for the
determination of cell viability in cell proliferation.
23. Line 510: why were the proteins crosslinked?
Thanks for your comments. This protocol describes the cross-linking of antibodies
to either protein A or G agarose beads using disuccinimidyl suberate (DSS)

Again a completely unnecessary statement because it is so described in the text. Binding to IgG by protein A or G is strong and sufficient for immunoprecipitation. Why was it crosslinked here?

24. Fig. S2: the panels are much too small.
Thanks for your comments. We have made changes in Fig. S2 accordingly in the
supplementary file.
25. Fig. S3: the figure is much too small. Nothing can be read.
Thanks for your comments. We have made changes in Fig. S3 accordingly in the
supplementary file.

see above

Author Response

Dear Reviewer, 

We are deeply sorry for the inconvenience caused and we take full responsibility for our negligence. During round one response to reviewer question, we did not understand some of the questions well enough, so that was the reason why  some questions were answered incorrectly. We are deeply sorry for our mistake and we hope for your understanding. Nevertheless, we have carefully revised the manuscript according to the reviewer comments and made an extensive modification on the revised manuscript, and we hope our responses are informative enough.  Thank you for your constructive comments concerning our article.
